DeepEMPR: coffee leaf disease detection with deep learning and enhanced multivariance product representation

Topal Ahmet atopal@itu.edu.tr 1
Tunga Burcu 1
Tirkolaee Erfan Babaee 2 3 4
1 Department of Mathematics Engineering, Istanbul Technical University , Istanbul , Turkey
2 Department of Industrial Engineering, Istinye University , Istanbul , Turkey
3 Department of Industrial Engineering and Management, Yuan Ze University , Taoyuan , Taiwan
4 Department of Mechanics and Mathematics, Western Caspian University , Baku , Azerbaijan
Alatas Bilal
Electronic publication date: 2024 Nov 13
Publication date: 2024
Volume: 10
Electronic Location ID: e2406
Received 2024 Jun 20; Accepted 2024 Sep 20
Copyright: ©2024 Topal et al.
Copyright year: 2024
Copyright holder: Topal et al.
License: This is an open access article distributed under the terms of the Creative Commons Attribution License, which permits unrestricted use, distribution, reproduction and adaptation in any medium and for any purpose provided that it is properly attributed. For attribution, the original author(s), title, publication source (PeerJ Computer Science) and either DOI or URL of the article must be cited.
License URL: https://creativecommons.org/licenses/by/4.0/

Keywords: Plant disease, Deep learning, Enhanced multivariance product representation, High dimensional model representation

Funding: Istanbul Technical University FHD-2024-45407 This work was supported by Istanbul Technical University within the scope of project number FHD-2024-45407. The funders had no role in study design, data collection and analysis, decision to publish, or preparation of the manuscript

==============================
Plant diseases threaten agricultural sustainability by reducing crop yields. Rapid and accurate disease identification is crucial for effective management. Recent advancements in artificial intelligence (AI) have facilitated the development of automated systems for disease detection. This study focuses on enhancing the classification of diseases and estimating their severity in coffee leaf images. To do so, we propose a novel approach as the preprocessing step for the classification in which enhanced multivariance product representation (EMPR) is used to decompose the considered image into components, a new image is constructed using some of those components, and the contrast of the new image is enhanced by applying high-dimensional model representation (HDMR) to highlight the diseased parts of the leaves. Popular convolutional neural network (CNN) architectures, including AlexNet, VGG16, and ResNet50, are evaluated. Results show that VGG16 achieves the highest classification accuracy of approximately 96%, while all models perform well in predicting disease severity levels, with accuracies exceeding 85%. Notably, the ResNet50 model achieves accuracy levels surpassing 90%. This research contributes to the advancement of automated crop health management systems.

Introduction

The agricultural industry serves as a cornerstone of many economies worldwide, profoundly impacting the growth and prosperity of communities (Pranto et al., 2021). As the global population continues to rise, ensuring sustainable and efficient agricultural practices becomes more critical than ever. However, plant diseases significantly threaten to agricultural productivity and sustainability by compromising crop quality and yield. Despite existing crop protection measures, plant diseases still lead to substantial yield losses, estimated at 20% to 40% (Richard, Qi & Fitt, 2022). To address the challenges posed by plant diseases, there is a growing imperative to develop artificial intelligence (AI)-based applications capable of accurately and promptly diagnosing these ailments. Such applications represent a forward-thinking approach to agricultural technology, offering smarter and more efficient means of diagnosing, monitoring, and managing plant health issues. By leveraging sophisticated deep learning (DL) models, these AI systems simplify the complex process of disease identification, empowering farmers and agronomists with timely insights into the specific pathogens affecting their crops. Recent advancements in AI have paved the way for automated systems designed to effectively manage crop health. These systems harness the power of DL to analyze vast amounts of data, thereby offering actionable recommendations for disease control and prevention. Consequently, the integration of AI technology holds tremendous potential in boosting crop yields and ensuring agricultural sustainability.

Recently, DL and machine learning (ML) have taken the forefront in the field of innovative AI solutions (Ghadi et al., 2023). Particularly in the field of image processing, ML has made significant strides through its sophisticated recognition and classification abilities. Traditional ML algorithms extract features from digital images and analyze these features to enhance their predictive capabilities. They can automatically identify patterns and relationships present in the image data. However, a limitation of traditional ML is its heavy reliance on the quality and relevance of manually engineered features. This reliance on manual feature engineering means that the success of ML algorithms in image processing is contingent upon the expertise of the humans who design and tune them.

On the other hand, DL, as a subfield of ML, possesses the ability to achieve high-performance modeling by utilizing the multi-layered structure of artificial neural networks. Convolutional neural networks (CNNs), as one of the DL architectures, are state-of-the-art tools for understanding the complexity and hidden motifs within image data (Ölçer, Ölçer & Sümer, 2023). Another aspect contributing to the popularity of CNNs is their ability to learn feature representations automatically, unlike traditional algorithms, which require manual feature extraction. Therefore, they are extensively used across various computer vision applications, such as classification, detection, and segmentation.

In recent years, research has increasingly focused on employing DL and ML techniques to recognize and diagnose plant diseases. This trend underscores the growing recognition of the potential of these advanced technologies in revolutionizing agricultural practices. In traditional ML-based studies, the system being proposed roughly consists of three key stages: preprocessing, feature extraction, and classification. Neelakantan (2021) collected a balanced dataset of 1,090 images to conduct research on the classification of diseases on tomato leaves. All images were annotated into the healthy class and four disease-related classes: septoria leaf spot, leaf curl, bacterial spot, and early blight. The performance analyses of supervised learning algorithms, including support vector machine (SVM) (Vapnik, 2013), K-nearest neighbor (Cover & Hart, 1967), decision tree (DT) (Quinlan, 1986), random forest (RF) (Breiman, 2001), and naive Bayes(NB), were evaluated based on the accuracy metric. Experimental findings indicated that RF was the top-performing classifier, achieving an accuracy of 89%. Ramesh et al. (2018) used the histogram of oriented gradient (HOG) (Dalal & Triggs, 2005) for the feature extraction phase in the binary plant disease classification and then implemented the RF, an ensemble classifier, during the training process. Prajapati, Shah & Dabhi (2017) designed a system that leverages the techniques of ML and image processing to enable the automated identification of diseased rice leaves. In the preprocessing step, the image backgrounds were removed using the S component of the image in the HSV domain. To eliminate bias during feature extraction, the diseased, non-diseased, and background segments were clustered using K-means (MacQueen, 1967). They then derived the features related to color, texture, and shape from those segments identified as diseased. During the classification step, the researchers experimented with three SVM models, each equipped with a different number of features, to study the impact of the parameters. Furthermore, ML research focused on different plant species, such as corn (Panigrahi et al., 2020), grape (Kaur, Pannu & Malhi, 2019), potato (Singh & Kaur, 2021), and rice (Ahmed et al., 2019), offers a broad spectrum of insights.

Mohanty, Hughes & Salathé (2016) performed a study on the recognition of 26 plant diseases using a publicly available database containing leaf images from 14 different plant species. By combining four main groups—deep learning architectures, training mechanisms, dataset types, and training-test set distributions—in various ways, they created and tested 60 different experimental configurations. Their results exhibited exceptional performance, with success rates reaching up to 99.35% in automated identification. A similar research was conducted by Ferentinos (2018) on a larger number of diseases and different plant species. Its strength was that it does not consider only laboratory-condition images but also includes real-condition images in the experimental phase. The results were exceptionally promising as the most successful model architecture attained a success rate of 99.53%. Too et al. (2019) adopted deep CNNs to create a model for recognizing plant diseases through leaf images. They built their model to identify 38 different classes including images of diseased and healthy leaves from 14 different plants. Their results showed that DenseNets were able to efficiently detect leaf diseases and classify test data with an accuracy of 99.75%. Agarwal et al. (2020) suggested a DL model that trains on tomato leaf images sourced from the Plant Village database (Too et al., 2019). The proposed architecture consists of three convolutional layers, with a max-pooling layer after each, and then two fully connected layers. The authors noted a significant finding that their model exhibited superior performance when compared to well-known CNN models like VGG16 (Simonyan & Zisserman, 2014), InceptionV3 (Szegedy et al., 2016), and MobileNet (Howard et al., 2017). Nafi & Hsu (2020) highlighted the minority of samples in the infected class, addressing the issue of class imbalance. In a comparative study by Sujatha et al. (2021), they indicated that DL techniques surpassed ML techniques in terms of classification accuracy. They also pointed out that among the tested models, RF had the lowest classification accuracy at 76.8%, while VGG16 achieved the highest at 89.5%. Shrivastava et al. (2019) presented an approach integrating ML and DL, where AlexNet (Krizhevsky, Sutskever & Hinton, 2012), a deep neural network, acted as the feature extractor, and the SVM acted as the classifier. The entire dataset was partitioned into training and testing sets at varying ratios, specifically 80%–20%, 70%–30%, and 60%–40%. In the experimental study, it was outlined how the different data partitions impacted the performance of the predictive model. Esgario, Krohling & Ventura (2020) made a major contribution to the coffee farming industry with automated disease diagnosis. They worked on both disease classification and disease severity classification. Their results showed that they almost perfectly detected the disease in coffee plants, but did not meet expectations in determining the disease’s severity.

Motivation for the study

Plant diseases present a major risk to agricultural productivity and sustainability by adversely affecting crop quality and yield. Despite current crop protection measures, plant diseases continue to cause significant yield losses. Furthermore, traditional methods for identifying leaf diseases rely heavily on manual inspection and expert knowledge, which can be time-consuming, costly, and prone to error. Additionally, existing automated techniques often struggle with accuracy due to the complex and variable nature of disease symptoms. Despite advancements in ML and computer vision, there remains a significant gap in developing robust, scalable, and accurate methods for disease identification.

Motivation for the choice of methods used

Over the past decade, CNNs have been commonly used to better process visual information. CNNs outperform traditional methods in visual information processing due to their ability to automatically learn hierarchical features from data. Furthermore, enhanced multivariance product representation (EMPR) is a technique for reducing the dimension by representing an N-direction tensor in the form of lower-dimensional tensors and specific support vectors. At the end of the EMPR process, the image data is separated into its high and low-frequency components. These components can be combined according to the task’s requirements to better represent the original images for classification, detection, and similar processes.

Contributions

In this work, we advance the research on coffee leaf disease detection by utilizing deep CNN architectures. Our algorithm integrates two mathematical methods introduced in recent literature, EMPR and high-dimensional model representation (HDMR). HDMR represents high-dimensional functions as combinations of lower-dimensional functions, such as univariate and bivariate functions, along with their interactions. It is a powerful tool for analyzing high-dimensional datasets, simplifying the analysis process, and revealing underlying relationships and interactions within the data. EMPR, derived from the HDMR framework, is specifically designed for representing multivariate functions and tensor-type datasets (Şen & Tuna, 2023). It extends the capabilities of HDMR to handle complex multivariate functions and adapt to data structures characterized by multiple dimensions or variables. EMPR achieves this by employing a more specific decomposition approach, enhancing the representation of high-dimensional data in terms of simpler, less-variate structures (Tuna et al., 2020; Tuna & Tunga, 2013). The use of these mathematical methods has enabled us to achieve superior results in coffee leaf disease detection compared to previous studies (Esgario, Krohling & Ventura, 2020).

Our algorithm is founded on the creation of a new channel for each original image. This involves incorporating a series of EMPR components obtained from the EMPR method (Tunga & Demiralp, 2010), which is utilized in various fields in the literature (Korkmaz Özay & Tunga, 2022; Tunga, 2015). The created channel allows the hybridization of useful details from both high-frequency and low-frequency components. After creating the channel, a contrast enhancement process is performed using an HDMR-based contrast enhancement method (Tunga & Kocanaogullari, 2017). Following this, the enhanced channel is added to the original three-channel image, converting the baseline dataset into a dataset of coffee leaves with four channels. CNN architectures are applied to this dataset to accomplish the disease classification and stress severity estimation tasks.

Structure of the manuscript

The remainder of this paper is divided into five main sections. The second section reminds the mathematical background of EMPR concepts. The third section introduces the dataset under consideration. The fourth section elaborates on the methodology proposed in this study, providing detailed insight into the steps that comprise our investigative approach. The fifth section presents the experimental study and its results. The last section includes concluding remarks on the paper.

Enhanced Multivariance Product Representation

EMPR is a technique for reducing the dimension by representing an N-direction tensor in the form of lower-dimensional tensors and specific support vectors. Employing the EMPR method, the representation of the N-dimensional tensor structure X reveals itself in the following manner: (1) Xi1…iN=X0 ∏j=1Nsijj+∑j1=1NXij1j1 ∏j=1j≠j1Nsijj+∑j1,j2=1j1<j2NXij1,ij2j1j2 ∏j=1j≠j1,j2Nsijj+⋯+Xij1ij2...ijNj1j2...jNij=1,2,…,nj,j=1,2,…,N.

This expansion involves a finite set of summations. X0 is a constant, Xj1 represents one-way tensors (or vectors), and Xj1j2 corresponds to two-way tensors, namely matrices. The components on the right-hand side can be viewed as tensors with an increasing number of dimensions.

The entities s(j) in the expansion are referred to as support vectors. The primary objective of this approach is to determine the fundamental structures of EMPR components by leveraging support vectors, with the aim of representing multivariate data with fewer variate components.

To uniquely obtain the components in the expansion, vanishing conditions are given through the support vectors: (2) ∑ijl=1njlWijljlsijljlXij1…ijkj1…jk=0l=1,2,…,k,k=1,2,…,N,

where (Wijj)s are weight vectors and these weight vectors satisfy the normalization conditions given below: (3) ∑ij=1njWijj=1,∑ij=1njWijjsijj2=1j=1,2,…,N.

The constant component of the EMPR, X0, is uniquely determined based on the conditions mentioned in Eq. (4): (4) X0= ∑i1=1n1⋯∑iN=1nN∏k=1NWikksikkXi1…iN.

The univariate components represented by Xj are computed using Eq. (5) by excluding the pertinent direction: (5) Xijj= ∑i1=1n1⋯∑ij−1=1nj−1 ∑ij+1=1nj+1⋯∑iN=1nN∏k=1j≠kNWikksikkXi1…iN−X0sijjij=1,2,…,nj,j=1,2,…,N.

The EMPR’s bivariate components are calculated using Eq. (6) in a similar manner, excluding the two related directions: (6) Xij,ikj,k= ∑i1=1n1⋯∑ij−1=1nj−1 ∑ij+1=1nj+1⋯∑ik−1=1nk−1 ∑ik+1=1nk+1⋯∑iN=1nN∏k=1j≠kNWikksikkXi1…iN−Xikksijj−Xijjsikk−X0sijjsikkj,k=1,2,…,N,ij=1,2,…,nj,ik=1,2,…,nk.

Other high-variate terms in the expansion are determined using the same philosophy.

Identifying the support vectors in the EMPR method is crucial as it directly impacts the method’s performance (Tunga & Demiralp, 2010; Tunga, 2015). While there are no constraints on how support vectors are evaluated, in this study, we chose the support vectors by normalizing the directional averages of the dataset below. The calculation of support vectors is thoroughly outlined in Eq. (7). For more detailed information about support vectors, readers may refer to Tuna et al. (2020), Şen & Tuna (2023), and Tunga (2015). (7) sijj=∑i1=1n1⋯∑ij−1=1nj−1 ∑ij+1=1nj+1⋯∑iN=1nNXi1…iN∑ij=1nj∑i1=1n1⋯∑ij−1=1nj−1 ∑ij+1=1nj+1⋯∑iN=1nNXi1…iN212.

Dataset Description

Our study focuses on the recognition of diseases in coffee leaves. We use the Arabica Coffee Leaves dataset (Esgario, Krohling & Ventura, 2020), which is a collection of high-resolution images showcasing various leaf conditions. It equips us with the means to investigate and identify the severity levels and various types of diseases affecting coffee plants. Our objective is to improve the accuracy of disease diagnosis, thereby contributing to the efficient management and treatment in coffee farming.

The leaf images are captured using smartphones at various times throughout the year in Marechal Floriano County, located in the state of Espírito Santo, Brazil (Esgario, Krohling & Ventura, 2020). The dataset comprises a total of 1,685 images, categorized into two primary classes: one representing healthy images and the other representing diseased images. Each image has dimensions of 512 pixels in height and 256 pixels in width. It features three color channels providing color information and is represented in the RGB color space. We created two datasets from the original: one to address recognizing biotic stress and the other to assess stress severity. Diseased leaf images are annotated into biotic stress categories, such as leaf miner, rust, brown leaf spot, and Cercospora leaf spot. They are also labeled with different levels of stress severity, including very low, low, high, and very high. Figures 1 and 2 show example images from each class of biotic stress and stress severity, respectively, except for the healthy class.

Figure 1 A sample image of each type of disease affecting coffee plant leaves.

Figure 2 A sample image from each disease severity class.

We split the datasets into training, validation, and test sets, for model training and evaluation. Exactly 70% of the data is designated for the training set, which is used to train our models. A 15% portion of the data is allocated to the validation set for model selection. This validation set serves as a checkpoint to assess the model’s performance during training, allowing us to make essential adjustments without impacting the integrity of the test data. It is also crucial for identifying and rectifying overfitting, where a model performs well on training data but poorly on unseen data. Finally, the remaining 15% of the data is reserved for the test set, used exclusively for evaluating the model’s performance. The data division process has been conducted under the guidance of the reference research (Esgario, Krohling & Ventura, 2020).

The number of images in each class for the datasets on biotic stress and stress severity is presented in Table 1. The table clearly shows that the sample distribution among the classes in both datasets is not uniform, leading to an imbalance between classes with one class containing a greater number of samples compared to the others. From a quantitative perspective, when the number of instances in the majority class (class with a greater number of examples) is divided by the number of instances in the minority class (class with fewer examples), imbalance ratios of 16.5 for severity and 3.61 for biotic stress are observed. This means that the biotic stress dataset is partially imbalanced, and the severity dataset is highly imbalanced (Topal & Amasyali, 2021). The high imbalance ratio makes it challenging for the classifier to learn the examples from the minority class. Therefore, it is required to use appropriate performance metrics that are more sensitive to the minority class.

Table 1 Dataset overview.

Biotic stress	# of images	Stress severity	# of images	
Brown leaf spot	348	Healthy	272	
Cercospora leaf spot	147	High	101	
Healthy	272	Low	332	
Leaf miner	387	Very high	56	
Rust	531	Very low	924	

Proposed Methods

The main focus of the study is to develop approaches that will enhance the generalization and learning capacity of the model during the preprocessing stage. An additional channel based on the EMPR will be added to the images. This aims to examine the impact of using more features. Additionally, the contrast of this channel, which will be added to the original images, will be improved to various degrees based on high-dimensional model representation, and the effect of this enhancement will also be researched. The steps to be applied in the study to investigate the impact of these criteria are listed below:

1. The dataset to be used in the study is split into training, validation, and test sets in specific proportions. This splitting is necessary for training the model, validating it, and testing the results.

2. During the preprocessing stage, a method based on EMPR is used to add a new channel to each image. With this method, each image will become four-channel by adding a new channel to the original three-channel image. Additionally, the contrast of the added channel is improved at different levels.

3. Thus, a training set is generated for each level of contrast enhancement.

4. CNN models are trained separately on each generated training set.

5. The performance of the trained models is investigated on the validation set, and the model showing the best performance (i.e., the one with the least loss value) for each level of contrast enhancement is selected.

6. Final performance analyses of the best-performing models are conducted on the test set using various metrics such as accuracy, precision, recall, and F1-score. These analyses are carried out to evaluate the generalization ability and performance of the models.

7. Additionally, the ability of the models to distinguish between diseased and healthy classes is investigated using ROC-AUC curves. These evaluations help to demonstrate the classification performance of the model in more detail.

Preprocessing stage

In image classification tasks, the effectiveness of the results largely depends on the robustness of the preprocessing steps applied to the raw data. In our study, the proposed pipeline for preprocessing stages is illustrated in Fig. 3 and additional details are presented through the pseudocode available in Algorithm 1 . Our steps prepare the raw image data for further analysis, improving features essential for classification. This approach enables classification algorithms to concentrate on the most important attributes of the data, a key factor in achieving high performance.

Figure 3 Preprocessing steps.

_______________________ Algorithm 1 Preprocessing Stage__________________________________________________________ Require: I← An RGB Image Ensure: Four Channel Image   1:  begin  2:       Find the height (n1), width (n2), and depth (n3) of I.   3:       Compute optimum support vectors s(1) i1  , s(2) i2  , and s(3) i3  .   4:       Compute   constant   (X(0)),   univariate   (X (1) i1   ,X (2) i2   ,X (3) i3   ),   bivariate      (X (1,2) i1,i2 ,X (2,3) i2,i3 ,X (1,3) i1,i3 ), and trivariate (X (1,2,3) i1,i2,i3) EMPR terms of I.   5:       Construct EMPR-based image G.   6:       Convert G to the grayscale image J(g).   7:       Compute constant, univariate, bivariate HDMR components of J(g).   8:       for i = 1 → α do  9:            Add the univariate and bivariate components to the J(g). 10:       end for 11:       Put the enhanced grayscale image as a new channel behind the I. 12:  end Notes: ⊳ α : degree of contrast enhancement ⊳  i1 = 1, 2,...,n1, i2 = 1, 2,...,n2, and i3 = 1, 2,n3 = 3______________________________________________

The pipeline begins with the computation of EMPR components from the original RGB image, which is represented by a 3-dimensional tensor. In EMPR frameworks, the terms Xi1,i21,2,Xi1,i31,3,Xi2,i32,3, and Xi1,i2,i31,2,3 can be categorized as high-frequency components, and the terms X0,Xi11,Xi22, and Xi33 can be categorized as low-frequency components (Korkmaz Özay & Tunga, 2022). EMPR-based image is derived through Eq. (8): (8) G=Xi33si11si22+Xi1,i31,3si22+Xi2,i32,3si11+Xi1,i2,i31,2,3i1=1,2,…,n1,i2=1,2,…,n2,andi3=1,2,n3=3,

where Xi33,Xi1,i31,3,Xi2,i32,3 and Xi1,i2,i31,2,3 denote the EMPR components; si11 and si22 denote the support vectors; and n1, n2, and n3 represent the height, width and depth information of the original image, respectively.

It is not constructed to include solely low-frequency or high-frequency terms; it is designed to take advantage of the beneficial properties of both. High-frequency terms Xi1,i31,3,Xi2,i32,3, and Xi1,i2,i31,2,3 are used to better capture the edges, fine details and textural information in the image. On the other hand, since high-frequency terms are easily affected by noise, and low-frequency terms are robust against noise, omitting the term Xi1,i21,2 and using the low-frequency term Xi33 relatively reduces the noise in the resulting image and offers a gradual transition of colors. The key aspect of using high-frequency EMPR components predominantly is that the high-frequency details significantly improve the discrimination of features, aiding in more accurate and refined object detection, classification, and analysis. Furthermore, the superposition of EMPR terms incorporating the {i3}-th position provides clearer visual differentiation between diseased and non-diseased regions in leaf images, as these terms emphasize color information well.

Following the EMPR construction, the image undergoes conversion to grayscale by applying the weights 0.2989 for the red channel, 0.5870 for the green channel, and 0.1140 for the blue channel. This well-known conversion formula is provided in International Telecommunication Union (2011): (9) Jg=0.2989GRed+0.5870GGreen+0.1140GBlue,

where GRed,GGreen, and GBlue correspond to the red, green, and blue channels of the image G, respectively.

This conversion reduces the image from three color channels (red, green, and blue) to a single channel representing intensity. The pipeline culminates with the creation of an enhanced image that is not only visually more striking but also carries a greater informational load. To produce the enhanced image, the univariate and bivariate HDMR components of the grayscale image are first computed, and then they are added to the grayscale image at a specified level. Finally, this enhanced image is combined with the original RGB data to create a four-channel image. This process integrates both image construction based on EMPR components and the contrast enhancement provided by HDMR.

Figure 4 shows a comprehensive illustration of the entire process in the pipeline on an example image. In this case, the alpha value for contrast stretching is set to 3.

Figure 4 Visualization of preprocessing steps on an example image.

Classification stage

Over the past decade, the field of CNNs has experienced remarkable evolution and innovation. A major factor in this development is the advent of CNN architectures, each with unique design and functionalities. Many of these architectures gained prominence through their success in the ImageNet Large Scale Visual Recognition Challenge (ILSVRC), a benchmark competition in the field. Even with their varying characteristics, these architectures all aim to better process visual information. Additionally, while some architectures may be widely used, the performance of one architecture over another can change depending on the dataset being used. In this study, we chose the AlexNet and VGG16 architectures, along with ResNet50, which is known for its strong results in (Esgario, Krohling & Ventura, 2020).

AlexNet: In 2012, Alex Krizhevsky and his team introduced a CNN model that was more sophisticated than LeNet and won the challenging ILSVRC in visual object recognition. It exhibited a high performance and achieved a lower error rate compared to existing models in the competition. The architecture of AlexNet is composed of eight layers in total and it accepts input images that are size of (224,224,3). The first five layers are specialized for convolution operations, and designed for feature extraction. These layers make use of a combination of max pooling and rectified linear unit (ReLU) activation functions to augment the learning process. Following these, there are three fully connected layers, which integrate the learned features for classification. Dropout is crucial in the fully connected layers to mitigate the problem of overfitting. The role of dropout here is to randomly disable a portion of neurons during the training, thus ensuring that the network does not become too dependent on any particular group of neurons.

VGG16: The VGG16 architecture was conceptualized and introduced by Karen Simonyan and Andrew Zisserman from the Visual Geometry Group Lab. In the 2014 ILSVRC competition, which was organized into two categories—object recognition and image classification—they won first place in object recognition and second in image classification. It accepts an input image of dimensions 224 × 224 pixels, with three color channels. The architecture is deep, comprising sixteen layers. The entire architecture has five groups of cascaded convolution layers. After each group of convolution layers, there is a max-pooling layer, which is implemented with a stride of (2,2), aiding in reducing the spatial size of the representation and in turn, the computational load. In the convolutional layers, receptive filters of different numbers, 64, 128, 256, and 512, are used. All these filters share a kernel size of (3,3). The final three layers are fully connected, with the first two layers having 4,096 neurons each. The last layer contains the number of neurons corresponding to the total number of classes in the classification task. The ReLU activation function is applied throughout the convolutional and hidden layers, while the softmax activation function is used in the last fully connected layer to compute the class probabilities. However, managing the VGG16 can be challenging due to its extensive structure, comprising approximately 138 million parameters.

ResNet50: ResNet50 is a version of the well-known ResNet architecture, which is short for “Residual Network”. This deep architecture introduced a novel concept known as ‘residual connections’. In a ResNet, the fundamental building block is the residual block, where the input to the block is connected to the output of the block, a connection known as the ‘residual connection’. The key feature is that it allows the layers to learn the residual function that maps the input to the target output, rather than learning the complete mapping from input to output. This makes it easier to optimize and train very deep networks.

The ResNet50 model starts with a convolutional layer, followed by batch normalization, and then applies a ReLU activation function. It continues with four major sets of residual blocks. Each set comprises multiple residual blocks, and each block is made up of several convolutional layers followed by batch normalization and a ReLU activation function. After the residual blocks, global average pooling is typically used to minimize the spatial dimensions of the feature maps. The network ends with a fully connected layer that performs the final classification.

The transfer learning approach is used to increase the efficiency of the training and to achieve faster results. We utilized the pre-trained CNN architectures trained on the ImageNet database. Several modifications are introduced to these pre-trained models:

• The number of units in the last linear layer is adjusted to match the number of classes in our dataset.

• Since an additional channel is added to each RGB image, the number of input channels in the first convolutional layer configuration is increased from 3 to 4.

• The network weights associated with this newly added channel are set to be the mean of the weights from the existing three layers.

During the training, no layers are frozen, that is, we allowed every layer of the network to be trained and adjusted. This is implemented to ensure that the network can adapt more comprehensively to the specific requirements of our dataset and tasks.

There are two jobs in the dataset: disease class prediction and stress level estimation. The single-task CNN and the multi-task CNN are two common methodologies we might adopt. Their fundamental difference lies in the configuration of their feature extraction layers. The single-task approach uses separate convolution and pooling layers dedicated to each task, whereas the multi-task approach uses shared convolution and pooling layers across tasks. We have chosen to pursue a single-task learning approach. This strategy is centered on focusing our computational resources and design efforts on one specific task and supports a more targeted and efficient learning process on our dataset.

We determined an optimal set of hyperparameters to drive our DL models. Our choices included stochastic gradient descent (SGD) as the optimizer, with cross-entropy as the loss function. Training occurred over 30 epochs, each processing a batch size of 32 images. The learning rate is initiated at 0.001 and implemented a scheduled decrease by a factor of 1/2 every 10 epochs to fine-tune the learning process. Additionally, momentum is set to 0.9 for efficient convergence. To prevent overfitting and promote generalization in our model, we applied a weight decay of 0.0005.

Dealing with small and imbalanced datasets brings about primary concerns related to biases and the risk of overfitting. Data augmentation methods offer a powerful solution to address these issues by artificially expanding and diversifying the training dataset. In image classification, augmentations like rotation, flipping, and brightness adjustments help create a more balanced representation of different angles, orientations, and lighting conditions. These transformations mimic real-world variations and infuse variability into the training data. In our approach to data augmentation, a variety of standard transformations are applied to produce synthetic images using the original training dataset. This included implementing random horizontal flips and random vertical flips, which introduce changes in orientation and mirroring. Furthermore, random rotations are integrated to simulate various viewpoints. To further enhance the diversity of color variations in our data, random adjustments in brightness, contrast, and saturation are performed.

The diagram in Fig. 5 illustrates an overview of our image classification task. The capsule-shaped section presents the sequence of operations that occur during a single epoch within a model training cycle. Firstly, a set of augmented training images is fed into the pre-trained CNN architecture. Once the model is trained, it is evaluated using a validation set. During the performance evaluation of the trained model, the validation loss is computed and saved. This process continues until the maximum number of epochs is completed. Then, the model with the lowest validation loss across all epochs is recognized as the best-performing model. Finally, the selected model is evaluated on a testing set, with its performance in classification being measured using metrics like accuracy, precision, recall, and the F1-score.

Figure 5 Flowchart of the classification process.

Experimental Results

Performance metrics

Performance metrics are essential in evaluating and guiding the development of ML and DL models. These benchmark functions act as quantifiable indicators of a model’s performance, allowing practitioners to impartially evaluate the extent to which their models are meeting the desired objectives. Fundamentally, these metrics are instrumental in enabling well-informed decision-making processes during the critical phases of model selection and tuning. They assist in the precise identification of the most appropriate algorithms and parameter configurations for a specific task. In real-world applications, these metrics offer valuable insights to stakeholders regarding the reliability and effectiveness of the models contributing to informed decision-making. They also play a key role in the analysis and assessment of model behavior, permitting the identification and rectification of critical issues, including but not limited to overfitting, underfitting, and biases.

In our study, various performance metrics (Bishop & Nasrabadi, 2006) were employed to demonstrate the effectiveness and success of our methodology. As indicated in ‘Dataset Description’, our datasets are partially imbalanced and highly imbalanced. Although the accuracy metric is not typically useful for imbalanced datasets, it was utilized to demonstrate the superiority of our results compared to the reference study (Esgario, Krohling & Ventura, 2020). Furthermore, we chose the metrics of recall and precision, which were included to allow for a more accurate assessment of a model’s performance in imbalanced datasets and to enable a direct comparison with the mentioned research. However, precision is insensitive to false negatives, which means it can provide an overly optimistic assessment when many relevant instances are missed. Conversely, recall is insensitive to false positives, potentially leading to a misleading sense of performance if many irrelevant instances are incorrectly classified as positive. We addressed these limitations in precision and recall by employing the F1-score. It harmonizes both and provides a more balanced evaluation. Moreover, due to the complexity and depth of the models, it is often necessary to use additional metrics. Therefore, we employed the area under the receiver operating characteristic (ROC) curve (Fawcett, 2006), known as AUC-ROC, to measure a model’s ability to distinguish between classes.

The metrics referred to are computed through the confusion matrix, which is depicted in Fig. 6. The confusion matrix provides detailed information on true positives (TP), false positives (FP), true negatives (TN), and false negatives (FN), and it also suggests deeper insights into how well a classification model is performing. Accuracy quantifies the ratio of correct predictions out of total predictions and is formulated by Eq. (10): (10) Accuracy=TP+TNTP+FN+FP+TN.

Precision measures the proportion of true positive results among all positive predictions. This metric is necessary for understanding how successful the model is in correctly identifying positive cases. It is defined by Eq. (11): (11) Precision=TPTP+FP.

Recall, also known as sensitivity, evaluates the model’s ability to identify all relevant instances, indicating the percentage of true positives that are correctly identified and is defined by Eq. (12): (12) Recall=TPTP+FN.

The F1-score is the harmonic mean of precision and recall. It offers a single metric that helps to convey the overall effectiveness of a model in classifying data correctly, balancing the trade-offs between false positives and false negatives. It is represented by Eq. (13): (13) F1-Score=2×Precision×RecallPrecision+Recall.

Figure 6 Confusion matrix.

Results and discussions

All experiments in this study are conducted using the popular open-source ML library PyTorch. The Intel Iris Xe Graphics is used as the GPU. The hardware components feature an 11th Gen Intel® Core™ i5-1135G7 CPU with a base clock speed of 2.40 GHz.

We experimented with different contrast enhancement levels. For the biotic stress dataset, the contrast enhancement levels are chosen as α ∈ {1, 2, 3}. For the stress severity dataset, a slightly narrower range of contrast enhancement levels is used, with α ∈ {1, 2}. Numerical results of the CNN models are presented in two tables. Table 2 details the results pertaining to the biotic stress dataset, while Table 3 presents the model results on the stress severity dataset. In both tables, results are organized according to the different levels of contrast enhancement applied. This shows a clear view of how contrast levels influence model performance. The best result for each evaluation metric is highlighted in bold font. Furthermore, the graphical results of CNN architectures in relation to each performance metric are depicted in Figs. 7A, 7B, 7C, and 7D obtained from the biotic stress dataset, and Figs. 8A, 8B, 8C, and 8D, obtained from the severity dataset. These graphs show how the models’ success differs from one another in related levels of contrast enhancement.

Table 2 Biotic stress dataset results.

	Biotic stress dataset	
Metrics:	Accuracy
%	Precision
%	Recall
%	F1-Score
%	
α	1	2	3	1	2	3	1	2	3	1	2	3	
AlexNet	94.86	94.47	94.07	93.92	94.55	95.29	91.79	89.84	89.51	92.71	91.54	91.44	
VGG16	95.65	95.65	94.07	96.62	96.65	92.42	92.60	93.38	90.36	94.15	94.73	91.24	
ResNet50	94.86	94.86	93.68	94.01	95.93	93.78	92.68	91.00	90.07	93.28	92.77	91.48	
Notes.

The best result for each evaluation metric is highlighted in bold.

Table 3 Stress severity dataset results.

	Severity dataset	
Performance
metrics	Accuracy
%	Precision
%	Recall
%	F1-Score
%	
α	1	2	1	2	1	2	1	2	
AlexNet	88.14	88.14	83.55	84.89	85.30	84.61	84.28	84.62	
VGG16	89.33	89.72	84.17	84.83	83.17	85.18	83.58	84.83	
ResNet50	86.96	90.11	83.21	86.35	74.30	79.51	77.06	81.84	
Notes.

The best result for each evaluation metric is highlighted in bold.

Figure 7 (A–D) Performance metrics comparison of CNN models across different contrast stretching levels for biotic stress dataset.

Figure 8 (A–D) Performance metrics comparison of CNN models across different contrast stretching levels for severity dataset.

Biotic stress dataset: VGG16 stood out as the architecture exhibiting the best results in all metrics. For VGG16, the best results in all metrics are produced at α = 2. Its precision scored above 96% at α levels 1 and 2, outperforming the other models. At every alpha level, it achieved a higher recall value than other models, suggesting a more reliable detection of samples. AlexNet demonstrated high consistency in accuracy metric across all α levels with over 94% accuracy, even though there was a slight decrease as α increased. Its precision and recall also remained robust, with only minor fluctuations. ResNet50’s performance is competitively comparable to AlexNet in terms of accuracy. It showed a notable increase in precision at α level 2, but its recall is remarkable, particularly at α level 1, which indicates a high rate of true positive detections. On the other hand, at the α = 3, performance metrics results are generally poorer than those observed at the two other α levels.

The impact of EMPR-based channel increment on the performance of CNN models is analyzed by contrasting it with the findings of study Esgario, Krohling & Ventura (2020). The addition of an extra channel led to significant improvements in the AlexNet model. It surpassed the accuracy, precision, and recall metrics of the reference study. Similarly, it had a positive effect on the VGG16 model, with a meaningful increment in the recall value. On the other hand, the accuracy and precision values of the VGG16 remained competitive with the results reported in Esgario, Krohling & Ventura (2020). However, our ResNet50 model fell behind the results of the reference study.

Stress severity dataset: Unlike the biotic stress dataset, no individual model achieves the best result in overall performance metrics. ResNet50 excelled in accuracy and precision, AlexNet in recall, and VGG16 in F1-score. At α = 1, VGG16 had the highest accuracy and precision with scores of 89.33% and 84.17%, respectively, while AlexNet led in recall and F1-score having a recall of 85.30% and an F1-score of 84.28%. Although ResNet50’s results are close to those of AlexNet and VGG16 in terms of accuracy and precision, it significantly underperformed in recall by a wide margin. This is evident in the F1-score and caused the gap to widen.

When α is set to 2, there is a general increase in performance across all metrics for each deep architecture. ResNet50 benefited more from the increased α parameter and hit the best results in accuracy and precision among others. Although ResNet50 has quite high accuracy for both α levels, its F1-score indicates that the model struggles to learn images due to the imbalanced structure of the dataset. VGG16 performed better than it did at α = 1, and it secured the highest ranking in both recall and F1-score. AlexNet, despite lagging slightly behind in accuracy and precision, still demonstrated considerable effectiveness with substantial scores across all metrics.

Channel incrementation marked a major contribution when compared to the findings of the reference research. All CNN architectures contributed to the overall accuracy within the range of 3–6 points. While Esgario, Krohling & Ventura (2020) is unable to reach accuracy above 90%, our ResNet50 exceeded this threshold. On the other hand, EMPR-based incrementation addressed the problem of poor precision and recall results, especially due to the extreme imbalance of the severity dataset. For both metrics, most of our models got results above 80%, while VGG16 is the only model in the reference study to exceed this limit. Furthermore, AlexNet delivered groundbreaking outcomes. It jumped by about 10 points on average in precision and about 12 points on average in recall. The models are ranked on these two metrics as follows: for precision metric improvement, it is AlexNet > ResNet50 > VGG16, and for recall metric improvement, it is AlexNet > VGG16 > ResNet50.

To determine which class is learned by the classifier with difficulty, ROC curves for each classification model are presented in Fig. 9. The area under the ROC curve provides information about how well the classifier responds to various classes. The AUC value falls within the range of 0 to 1. The higher the AUC, the better the model’s prediction. An AUC of 0.5 indicates a lack of discriminative power, while an AUC of 1 indicates perfect discrimination. A classifier with an AUC below 0.5 implies that it has performance worse than random guessing.

Figure 9 (A–D) ROC curves of each classification model for severity dataset.

The “class healthy” curves are very close to the top left corner with an AUC of approximately 1 for all three models, indicating an excellent performance in distinguishing the healthy class from the rest. Although “class very high” is a minority class, it is also easily recognized by the models. Both classes are well-separated from the other categories. “Class high” and “Class low” have the lowest AUC scores and display less confident predictive abilities. There may be some overlaps in the features with other classes, which reduces their distinctiveness. Increasing α from 1 to 2, AlexNet and ResNet50 recognized “class low” and “class high” images better. This change in α appears to have a positive impact on AlexNet and ResNet50, allowing them to accurately detect “class low” and “class high” images. However, as alpha increases, VGG16 struggles to recognize both kinds of images.

Conclusion

Detecting diseases in coffee leaves is essential for maintaining the sustainability of coffee cultivation. Accurate and timely diagnosis not only aids in the effective management of crop health but also significantly contributes to the prevention of widespread agricultural losses. Our proposed study introduced a pre-processing strategy that combines the HDMR and EMPR concepts, and several parameters were key to advancing the diagnosis of diseases and disease severity in coffee leaves. In the pre-processing phase, in addition to the R, G, and B channels, a robust channel is formed by mixing low and high-frequency EMPR components, and it is further reinforced with contrast improvements based on HDMR. Additionally, the performance of deep CNN architectures has been investigated across various contrast enhancement levels for both tasks.

For the purpose of experimentation, well-known CNN architectures such as AlexNet, ResNet50, and VGG16 are implemented by adopting a single-task approach with transfer learning. Furthermore, data augmentation techniques are used to avoid the lack of learning due to the presence of fewer examples in one class compared to others. The testing results revealed that VGG16 performed better than previous research in obtaining the most accurate diagnoses for diseases in coffee leaves. This success was attained by incorporating a new EMPR-based channel and applying contrast enhancement at the α = 2 level. When using the same parameter settings, but with any alpha level, AlexNet has significantly improved upon the disease identification results obtained in the reference study. Moreover, our work successfully addressed the challenge in stress severity estimation that the previous study could not handle and fell short of. All CNN architectures achieved not only higher precision and recall but also an accuracy of around 90%.

Although this study has demonstrated the effectiveness of the proposed method in enhancing the classification capability of deep architectures, several areas remain unexplored. One of the primary factors influencing classification success is the bias introduced by the background, which should be addressed in future research. Efforts could aim to adapt EMPR and HDMR to disease segmentation. Additionally, we believe that developing novel sampling techniques inspired by EMPR could help overcome the frequently observed between-class imbalance in such disease datasets.

Supplemental Information

Supplemental Information 1 Leaf Disease Identification and Severity Estimation Python Code

Additional Information and Declarations

Competing Interests

Author Contributions

Data Availability

Erfan Babaee Tirkolaee is an Academic Editor for PeerJ.

Ahmet Topal conceived and designed the experiments, performed the experiments, analyzed the data, performed the computation work, prepared figures and/or tables, authored or reviewed drafts of the article, and approved the final draft.

Burcu Tunga conceived and designed the experiments, performed the experiments, analyzed the data, performed the computation work, prepared figures and/or tables, authored or reviewed drafts of the article, and approved the final draft.

Erfan Babaee Tirkolaee conceived and designed the experiments, performed the experiments, analyzed the data, performed the computation work, prepared figures and/or tables, authored or reviewed drafts of the article, and approved the final draft.

The following information was supplied regarding data availability:

The data, algorithms and code are available at GitHub and Zenodo:

- https://github.com/ArticleCodeHub/DeepEMPR

- Topal, A. (2024). DeepEMPR. Zenodo. https://doi.org/10.5281/zenodo.13823450.

The code is available in the Supplemental File.

The data is also available at Figshare: Topal, Ahmet (2024). LeafData.zip. figshare. Dataset. https://doi.org/10.6084/m9.figshare.26060464.v1.

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
