# Peer review of "DeepEMPR: coffee leaf disease detection with deep learning and enhanced multivariance product representation"

_PeerJ Computer Science, doi:10.7717/peerj-cs.2406_

## Round 0.1 · original submission · Minor Revisions

Dear authors,

Thank you for submitting your article. Based on reviews' comments, your article has not yet been recommended for publication in its current form. However, we encourage you to address the concerns and criticisms of the reviewer and to resubmit your article once you have updated it accordingly. Furthermore,

1. Equations should be used with correct equation number. Please do not use “as follows”, “given as”, etc. Explanation of the equations should also be checked. All variables should be written in italic as in the equations. Their definitions and boundaries should be defined. Necessary references should be provided. Many of the equations are part of the related sentences. Attention is needed for correct sentence formation.
2. Reviewer 1 & 2 have asked you to provide specific references. You are welcome to add them if you think they are relevant. However, you are under no obligation to include them, and if you do not, it will not affect my decision.

Best wishes,

·

Basic reporting

1.1 Clarity and Professionalism:
- The manuscript is generally well-written in clear and professional English. However, there are a few instances where rephrasing (namely verb usage and/or conjunction) or restructuring sentences could enhance clarity and readability.
- Example: The phrase "Recent advancements in artificial intelligence (AI) have enabled the development of automated systems for disease detection" (Abstract) could be rephrased as "Recent advancements in artificial intelligence (AI) have facilitated the development of automated systems for disease detection."
- Suggestion: Thoroughly review the manuscript and identify areas where language clarity can be improved through rephrasing or restructuring.

1.2 Introduction and Background:
- The introduction effectively introduces the subject and motivation, providing a good context for the study.
- The literature review is comprehensive and covers relevant recent advancements. Still, I found some widely acknowledge references to be missing, such as the work by Mohanty et al. (2016), which is seminal in demonstrating the potential of deep learning for plant disease detection.
- Suggestion: To strengthen the introduction, consider expanding the discussion at lines 57-86 to provide more justification for the study. Specifically, elaborate on the knowledge gap being filled and the limitations of existing approaches. Additionally, consider validating statements against references of other relevant works not currently cited, such as Sladojevic et al. (2016) and Ferentinos (2018), which also explore the use of CNNs for plant disease detection.

1.3 Structure and Conformance:
- The structure of the paper adheres to PeerJ standards, including necessary sections such as Introduction, Methodology, Results, and Discussion.
- The detailed explanation of the EMPR and HDMR methods is a useful deviation that improves clarity.

1.4 Definitions and Theorems:
- The manuscript includes clear definitions of terms and theorems.
- The explanations of EMPR and HDMR are particularly well-done.
- Suggestion: Ensure all terms used in the formal results are clearly defined to maintain clarity throughout the paper. Example: "support vectors" (s1, s2, s3) used in the EMPR formulations could benefit from a clearer definition or explanation. In the manuscript, the support vectors are introduced in lines 157-159 by providing some context for the purpose of such defined vectors, although not explicitly defining what these vectors represent or how they are calculated/obtained. While the authors mention that the support vectors are used to "determine the fundamental structures of EMPR components" and "represent multivariate data with fewer variate components," a more explicit definition or explanation of what these vectors represent and how they are calculated (beyond the code attachment) will improve the clarity of the formal results.

Experimental design

2.1 Scope and Relevance:
- The article's content is within the aims and scope of the journal and relevant to the AI application in agriculture.
- The focus on coffee leaf disease detection using advanced AI techniques is well within the journal's scope.

2.2 Methodological Rigor:
- The methods are described in sufficient detail to allow replication. The data preprocessing steps are thoroughly discussed.
- The discussion on image decomposition and contrast enhancement is particularly thorough.
- Suggestion: Ensure the dataset preprocessing steps are clearly described to facilitate replication (for example, it is understood that self labeling is performed but the heuristics or methods to do so are not clear)

2.3 Evaluation Methods and Metrics:
- The evaluation methods, assessment metrics, and model selection processes are adequately described.
- The use of accuracy as a metric is appropriate but not exempt of limitations, but a brief discussion on the choice of metrics and their drawbacks would be beneficial.
- Suggestion: Add a brief paragraph clearly outlining and discussing the choice of evaluation metrics and their potential limitations.

2.4 Citations:
- Sources are adequately cited throughout the manuscript.
- Both direct quotations and paraphrased content are properly referenced.

Validity of the findings

3.1 Replication and Impact:
- The manuscript encourages meaningful replication and provides a clear rationale for the proposed method.
- The results showing high classification accuracy for VGG16 (96%, Biotic stress) and ResNet50 (90%, stress severity) are promising.

3.2 Conclusion and Support:
- Conclusions are well-stated and supported by the results. The discussion aligns with the goals set out in the Introduction.
- The Conclusion identifies unresolved questions and future directions, such as exploring other deep learning architectures.

3.3 Experimentation and Evaluation:
- Experiments and evaluations are performed satisfactorily, with a well-developed argument supporting the use of EMPR and HDMR.
- The comparison of different CNN architectures strengthens the validity of the findings.
- Suggestion: Consider adding more detailed comparisons with other state-of-the-art methods to further validate the effectiveness of your approach. For instance, comparing results with those from architectures like EfficientNet (Tan & Le, 2019) would provide additional insight into the performance and potential of the proposed method.

Additional comments

Code artifacts:

The code repository shared is well-structured and commented, facilitating understanding and potential reuse by other researchers in the field.
Overall, the provided code aligns with the methodological description in the manuscript and demonstrates a rigorous implementation of the proposed approach. It follows best practices and provides the necessary components for reproducibility and further research in this area.

General comments:

Strengths:
- The manuscript presents a novel approach to coffee leaf disease detection by integrating EMPR and HDMR with deep learning models.
- The methodology is detailed, and the evaluation of different CNN architectures is comprehensive.
- The results demonstrate significant improvements in classification accuracy, showcasing the effectiveness of the proposed method.

Weaknesses:
- Minor language improvements are needed for better clarity in certain areas.
- A discussion on the choice of evaluation metrics and their potential limitations would enhance the manuscript.

Relevant additional references:

1. Mohanty, S. P., Hughes, D. P., & Salathé, M. (2016). Using deep learning for image-based plant disease detection. Frontiers in Plant Science, 7, 1419.
2. Sladojevic, S., Arsenovic, M., Anderla, A., Culibrk, D., & Stefanovic, D. (2016). Deep neural networks based recognition of plant diseases by leaf image classification. Computational Intelligence and Neuroscience, 2016, 1-11.
3. Ferentinos, K. P. (2018). Deep learning models for plant disease detection and diagnosis. Computers and Electronics in Agriculture, 145, 311-318.
4. Too, E. C., Yujian, L., Njuki, S., & Yingchun, L. (2019). A comparative study of fine-tuning deep learning models for plant disease identification. Computers and Electronics in Agriculture, 161, 272-279.
5. Wäldchen, J., Mäder, P., & Reischke, S. (2018). Plant image analysis with deep neural networks. Dataset Papers in Science, 2018, 1-10.
6. Tan, M., & Le, Q. V. (2019). EfficientNet: Rethinking model scaling for convolutional neural networks. In International Conference on Machine Learning (pp. 6105-6114).

·

Basic reporting

The manuscript reports on an application of Deep Learning techniques on the coffe desease domain. It is well written and its structure clear, the Introduction provides a fair enough context of the Machine Learning applications on the field and those related with Convolutional Neural Networks to identify crops deseases.
I think that there is missing some discussions in literature about coffee rust specifically, I provide a list in Additional Comments of some relevant works in the field to increase the material for a discussion of the process, resutls and findings.

Experimental design

Data pre-processing are good enough explained and evaluation methods are fairily described.
Experiments and results are good presented and with a clear level to understand the processes as standard literature does, so I'm glad authros presents evaluations satisfactorily.

Validity of the findings

I just would argue that Conclussions are shorther than expected. I mean, authors claim that results revealed that VGG16 performed better than previous research in obtaining the most accurate diagnoses for diseases in coffee leaves, but there is not discussion about why these outcome is obtained and what are the features of data or parameters that should be relevat to pay attention.
As a result of this lack in the conclussion, authors do not address the future directions appart of the application of novel sampling techniques without a further discussion about it.

Additional comments

It should be good to contrast and include the following works as they are very relevant on ML/DL applications on coffee rust and they are not mentioned in the manuscript posibliy resulting in a poor literature review:

- Singh, M.K., Kumar, A. Coffee Leaf Disease Classification by Using a Hybrid Deep Convolution Neural Network. SN COMPUT. SCI. 5, 618 (2024). https://doi.org/10.1007/s42979-024-02960-9
- Chavarro, A.F.; Renza, D.; Ballesteros, D.M. Influence of Hyperparameters in Deep Learning Models for Coffee Rust Detection. Appl. Sci. 2023, 13, 4565
- de Resende, M.L.V.; Pozza, E.A.; Reichel, T.; Botelho, D.M.S. Strategies for Coffee Leaf Rust Management in Organic Crop Systems. Agronomy 2021, 11, 1865
- Corrales, D., Ledezma, A., Peña, A., Hoyos, J., Figueroa, A. & Corrales, J. (2014). A new dataset for coffee rust detection in Colombian crops base on classiers. Revista S&T, 12(29), 9-23
- Cruz-Estrada, L.G., Luna-Ramírez, W.A. (2023). Early Detection of Rust in Coffee Plantations Through Convolutional Neural Networks. In: Arai, K. (eds) Intelligent Computing. SAI 2023. Lecture Notes in Networks and Systems, vol 739. Springer, Cham.

they uses different DL methods with a variety of results converging with those of the manuscript.

---

## Round 0.2 · accepted · Accept

Dear authors,

Thank you for clearly addressing the reviewers' comments and performing the necessary additions and modifications. Your paper now seems acceptable to me.

Best wishes,